# miR-10a-5p Regulates the Proliferation and Differentiation of Porcine Preadipocytes Targeting the KLF11 Gene

**DOI:** 10.3390/ani14020337

**Published:** 2024-01-22

**Authors:** Wanfeng Zhang, Tianzhi Zhao, Xinyu Gao, Shuangji Ma, Tianye Gong, Yang Yang, Meng Li, Guoqing Cao, Xiaohong Guo, Bugao Li

**Affiliations:** College of Animal Science, Shanxi Agricultural University, Jinzhong 030801, China; zhangwanfeng123@126.com (W.Z.); 19725845405@163.com (T.Z.); gxy793169147@163.com (X.G.); 19834541517@163.com (S.M.); gongty0708@163.com (T.G.); yangyangyh@163.com (Y.Y.); 13994576150@163.com (M.L.); anniecao710502@aliyun.com (G.C.)

**Keywords:** miR-10a-5p, pig, preadipocyte, proliferation, differentiation, KLF11

## Abstract

**Simple Summary:**

The aim of this study was to explore the regulatory effects of miR-10a-5b on the proliferation and differentiation of porcine preadipocytes. Our results showed that miR-10a-5b targets KLF11, suppresses proliferation, and promotes differentiation in preadipocytes. Therefore, we concluded that miR-10a-5b plays a significant role in regulating the proliferation and differentiation of preadipocytes.

**Abstract:**

In the swine industry, meat quality, color, and texture are influenced by the excessive differentiation of fat cells. miRNAs have emerged as integral regulators of adipose development. This study delves into the influence of miR-10a-5b on the proliferation and differentiation of pig preadipocytes. Our findings reveal that miR-10a-5b is prevalent across various tissues. It hinders preadipocyte proliferation, amplifies the expression of adipogenic genes, promotes lipid accumulation, and, as a result, advances preadipocyte differentiation. We predict that KLF11 is the target gene of miRNA. A dual-fluorescence reporter assay was conducted to validate the binding sites of miR-10a-5b on the 3′UTR of the KLF11 mRNA. Results showed that miR-10a-5b targeted KLF11 3′UTR and reduced the fluorescence activity of the dual-fluorescent reporter vector. Our research also indicates that miR-10a-5b targets and downregulates the expression of both mRNA and the protein levels of KLF11. During the differentiation of the preadipocytes, KLF11 inhibited adipose differentiation and was able to suppress the promotion of adipose differentiation by miR-10a-5b. This underscores miR-10a-5b’s potential as a significant regulator of preadipocyte behavior by modulating KLF11 expression, offering insights into the role of functional miRNAs in fat deposition.

## 1. Introduction

Adipose tissue, the primary reservoir of the body’s energy, is pivotal in regulating energy, metabolism, and equilibrium. The differentiation of adipose tissue not only impacts the quality of livestock meat and carcasses but also has implications for human health, contributing to conditions such as obesity. Traits related to fat deposition exhibit genetic variations, qualifying them as quantitative traits. Identifying molecular markers that influence porcine fat deposition can expedite genetic advancements. Therefore, over the past decade, local and international research has intensely focused on adipose tissues and adipocytes, marking them as prominent areas of study [1,2].

In recent years, non-coding RNAs, encompassing miRNAs, lncRNAs, and circRNAs, have emerged as major players in governing the proliferation and differentiation of porcine preadipocytes. Specifically, miRNAs are endogenous non-coded small RNA molecules, typically 18~24 nt in length, and display conservation across species with distinct tissue-specific temporal and spatial patterns [3]. Recent research has illuminated the role of miRNAs in regulating various physiological functions, such as lipid and glucose metabolism [4]. These miRNAs can directly or indirectly influence adipose differentiation. For instance, miR-148a has been identified as an obesity biomarker that facilitates fat differentiation by inhibiting Wnt1 [5]. In contrast, miR-146b augments adipose differentiation by adjusting KLF7 [6]. Research indicates that miR-370 limits adipogenic differentiation in porcine preadipocytes [7], while miR-26b and miR-1908 have been found to suppress adipogenic differentiation [8,9]. miR-23b promotes the differentiation of porcine preadipocytes by targeting SESN3 and promoting the expression of ACSL4 [10]. miR-503 directly targets the musculoaponeurotic fibrosarcoma oncogene homolog K (MafK) in proliferating and differentiating preadipocytes to repress adipogenesis [11]. Studies have shown that miR-196b-3p and miR-450b-3p are novel key regulators with opposite roles in porcine adipogenesis [12]. However, the functions of many miRNAs in adipogenesis have not been reported yet.

Numerous studies have shown that miR-10a-5p is crucial for the development of illnesses related to fat metabolism [13]. Increasing miR-10a-5p levels inhibited proinflammatory gene expression in macrophages and promoted the differentiation of C3H10T1/2 cells into brown adipocytes. The treatment of mice with a miR-10a-5p mimic suppressed pro-inflammatory responses, promoted the appearance of new white adipocytes in gWAT, and improved systemic glucose tolerance [14]. miR-10a-5p is involved in oxidative stress, inflammation, and insulin signaling. miR-10a-5p also has a strong association with abdominal adipocyte differentiation [15]. In addition, MiR-10a-5p regulates adipogenesis in mice [16]. However, the effects of miR-10a-5p on porcine preadipocyte differentiation are currently unknown.

Krüppel-like factors (KLFs) belong to the expansive zinc finger protein superfamily and are central to various cellular processes, including cell proliferation, apoptosis, differentiation, and embryonic development. They can either promote or inhibit pre-adipocyte differentiation during adipogenesis [17]. The KLF11 gene, a part of the KLF family, exhibits remarkable conservation across species, from flies to humans [18]. In mammals, KLF11 has also been recognized as a key component in the transcriptional network governing adipose differentiation, adipogenesis, and the onset of obesity [19]. Functionally, KLF11 can activate the PPARα signaling pathway, enhancing fatty acid oxidation [20]. Moreover, it has the ability to suppress the expression of SREBP-1c and FASN in the mouse liver, therefore inhibiting lipid synthesis [21]. While the KLF11 gene has an established role in lipid metabolism, its influence on pig fat proliferation and differentiation remains uncharted.

The primary aim of our investigation is to elucidate the modulatory role of miR-10a-5p in the differentiation of pig preadipocytes. Bioinformatics technology prediction results showed that KLF11 is the target gene of miR-10a-5p. In the present study, a dual-luciferase reporter gene assay was conducted to explore the mechanism of miR-10a-5p in regulating the differentiation of porcine precursor adipocytes through KLF11. It provides a theoretical basis for studying the molecular mechanism of fat metabolism in pigs. The insights gained from our study provide a novel perspective and a foundational framework for refining meat quality through basic research.

## 2. Materials and Methods

### 2.1. Ethical Approval Declarations and Sample Collection

All animal-related procedures adhered to the Code of Ethics at the World Medical Association (Declaration of Helsinki) and were conducted in alignment with the standards set by the American Physiological and World Medical Association General. The methodologies employed were grounded in the guidelines of the College of Animal Science and Veterinary Medicine at Shanxi Agricultural University (Taigu, China). The experimental protocols underwent review and received approval from this institution. The Ethics Committee’s approval reference is SXAU-EAW-P002003.

Jinfen White pigs (castrated boars) were used as experimental animals in the present study, provided by Datong Pig Farm (Shanxi, China). Three 90-day-old Jinfen White pigs (castrated boars) were killed by electric shock and bloodletting. After that, the longissimus dorsi muscle, heart, spleen, kidney, liver, lung, stomach, and subcutaneous fat were collected. In addition, nine Jinfen White pigs (castrated boars) at 1, 90, and 180 days of age were selected, with 3 pigs per age. Their tissues were instantly frozen in liquid nitrogen and afterward kept in a −80 °C refrigerator.

### 2.2. Cell Culture and Differentiation Induction

Subcutaneous adipose tissue from the back and neck of 7-day-old piglets was harvested and transferred to sterile Petri dishes. This tissue underwent digestion with an equal volume of collagenase type I at 37 °C for 30 min on a shaking platform. To isolate the digested cells, steel mesh filters with pore sizes of 200 µm and 70 µm were utilized. The filtrated cells, after being washed with DMEM/F12 medium, were subjected to centrifugation at 1000 r/min for 10 min, repeated twice. Following this, the supernatant was discarded, and the resulting pellet was re-suspended in DMEM/F12 supplemented with 10% fetal bovine serum (FBS). These cells were then cultured in Petri dishes at 37 °C in a 5% CO_2_ environment. All procedures adhered to appropriate guidelines and regulations.

For adipogenesis, preadipocytes were cultured in 12-well plates using an induction medium [0.5 mM 3-isobutyl-1-methylxanthine (IBMX), 1 µM dexamethasone (DEX), 10 mg/L insulin (MDI)]. After 48 h, this differentiation medium was replaced with a maturation medium (MM) that consisted solely of 10 mg/mL insulin. From the fourth day onwards and at every subsequent two-day interval, the cells received nourishment from a medium enriched with 10% FBS (Thermo Fisher Scientific, Waltham, MA, USA).

Separately, 293T cells were cultured in complete medium. Upon achieving a 70% confluence, these cells underwent transfection using the Lipofectamine 3000 reagent (Thermo Fisher Scientific, Waltham, MA, USA). Six hours post-transfection, the medium was refreshed.

### 2.3. Oil Red O Staining

Oil Red O staining (ORO) staining was employed to assess preadipocyte differentiation. After 8 days of differentiation, the cells were rinsed twice with phosphate-buffered saline (PBS). Subsequently, they were fixed using formaldehyde for a duration of 10 minutes. Following another set of washes with PBS, the cells were treated with freshly prepared ORO for 15 min. Post-staining, the cells received another PBS wash and were then visualized and captured using a microscope (Leica, Hessen, Germany).

### 2.4. Quantitative Real-Time Polymerase Chain Reaction

Total RNA was isolated from the cells using the TRIzol reagent (Takara, Dalian, China). The PrimeScript RT Reagent Kit (Takara, Japan) was used to synthesize the cDNA of mRNA, while the miRNA 1st Strand cDNA Synthesis Kit (Vazyme, Nanjing, China) was used to synthesize the cDNA of miRNA. The amplification reactions were set up using the corresponding amplification primers in conjunction with the SYBR Green PCR Master Mix (Takara, Dalian, China), with each reaction having a total volume of 20 μL. The mRNA expression levels were normalized against the reference genes 18S or U6. Details regarding the primers utilized for qRT-PCR can be found in Table 1. The genome referenced for primer design was Sus scrofa (taxid:9823).

### 2.5. Transfection

For the miR-10a-5p studies, cells were transfected with either miR-10a-5p mimics, inhibitors, or the appropriate negative controls. GenePharma (Shanghai, China) synthesized the small-interfering RNA (siRNA) targeting KLF11. The KLF11 expression plasmid was constructed by integrating the amplified KLF11 cDNA fragments into an empty vector. The KLF11 interference (designated as si-KLF11) or negative-control (si-NC) preadipocytes were generated using siRNA. For KLF11 overexpression, the preadipocytes were constructed using the expression plasmid and named KLF11-overexpressing cells, with the control cells labeled as the vector. For transfection, cells were initially incubated in a serum-free medium for 2 h. Subsequently, either the plasmid or siRNA was introduced into the cells using Lipofectamine 3000 transfection reagent, in accordance with the manufacturer’s guidelines (Thermo Fisher Scientific, Waltham, MA, USA).

### 2.6. Cell Proliferation Assay

For the 5-ethynyl-2′-deoxyuridine (EdU) assay, the preadipocytes were seeded in 12-well plates. After being transfected for 48 h, the preadipocytes’ proliferation was assessed using the Cell Proliferation EdU Image Kit (Abbkine, China). Subsequently, images were captured with a fluorescence microscope (Leica, Hessen, Germany).

The preadipocytes were also uniformly seeded in a 96-well plate and incubated at 37 °C with 5% CO2 until they reached the appropriate confluency for transfection. The 96-well plates were then removed at intervals of 12 h, 24 h, 36 h, 48 h, and 60 h post-transfection. Following the manufacturer’s protocol, 100 µL of the original medium was carefully removed from each well and replaced with 100 µL of complete medium supplemented with 10% CCK-8 reagent (Dojindo, Kyushu Island, Japan). The plates were then returned to the incubator for an additional 3 h. After post-incubation, the samples were read on an enzyme-linked immunoassay instrument (BioTek, Winooski, VT, USA), measuring absorbance (OD value) at a 450 nm wavelength.

### 2.7. Flow Cytometric Analysis

The preadipocytes were collected 48 h post-transfection and subsequently washed twice with PBS. Then, the cells’ DNA was stained with propidium iodide (PI) solution (Solarbio, Beijing, China) and incubated at 4°C for 30 min. Flow cytometric analysis of the preadipocytes was performed using a flow cytometer (Becton, Dickinson and Company, Franklin Lakes, NJ, USA), which was set to an excitation wavelength of 488 nm and an emission wavelength of 530 nm.

### 2.8. Dual-luciferase reporter assay

The luciferase reporter assay was conducted using the 3′UTR of KLF11, which encompassed the wild-type or mutant miR-10a-5p target sequences. Subsequently, the constructed vectors were co-transfected into 293T cells, along with miR-10a-5p mimics or inhibitors. Following a 48-hour transfection period, cells were collected. The luciferase activity was quantified using the Dual-Luciferase Reporter Assay System Kit (Promega, Madison, WI, USA).

### 2.9. Western Blotting Analysis

Total proteins were extracted from the cells in the knockout and overexpression groups, as well as their respective control groups. For gel electrophoresis, 20 μg of protein from each group was loaded onto a sodium dodecyl sulfate-polyacrylamide gel (SDS-PAGE). After electrophoresis, the proteins were transferred onto a membrane. This membrane was then blocked and incubated successively with primary and secondary antibodies. The primary antibodies against KLF11, PPARγ, and aP2 were sourced from ABclonal (Wuhan, China), while the secondary antibody was acquired from LICOR. The final step involved analyzing the optical density values of the target band using the Odyssey FC NIR Protein Processing System (LI-COR, Lincoln, NE, USA).

### 2.10. Bioinformatics Analysis

The sequences for miR-10a-5p and the 3′UTR of the target gene were obtained from the NCBI database. Potential target genes for miR-10a-5p were predicted using three online tools: miRDB, Targetscan, and TangetMiner. An intersectional analysis of the results from these tools was conducted to identify consensus target genes.

### 2.11. Statistical Analysis

All data were analyzed using SPSS version 22.0 and are presented as means ± SEMs. The statistical significance was determined at a *p* value of < 0.05. Data visualization was performed using GraphPad Prism 7.0 software (GraphPad Software, San Diego, CA, USA).

## 3. Results

### 3.1. miR-10a-5p Expression Pattern in Pigs

In this study, the expression profile of miR-10a-5p was analyzed across different pig tissues. The results showed that miR-10a-5p was ubiquitously expressed in a variety of pig tissues. Interestingly, a pronounced expression of miR-10a-5p was observed in the lung and kidney. The adipose tissue also showed notable levels of miR-10a-5p expression (Figure 1A). The study aimed to further discern the temporal expression trend of miR-10a-5p during the developmental stages of subcutaneous adipose tissue in pigs. By extracting the total RNA at different growth and developmental time points, it was discerned that the expression of the miR-10a-5p gene peaked on day 180, while it was the least on day 0 in the subcutaneous adipose tissue of pigs (Figure 1B). In addition, the expression of miR-10a-5p varied throughout the adipogenic differentiation process of porcine precursor adipocytes, with the highest expression on day 0 of differentiation (Figure 1C). These observations suggest a potential regulatory role for miR-10a-5p in the early stages of adipogenic differentiation in pig preadipocytes.

### 3.2. miR-10a-5p Inhibition of Porcine Preadipocyte Proliferation

To elucidate the impact of miR-10a-5p on porcine preadipocyte proliferation, we utilized a range of assays, including qRT-PCR, CCK-8, EdU, and flow cytometry. Our data showed an approximately 15-fold increase in the transfection efficiency of miR-10a-5p when preadipocytes were treated with miR-10a-5p mimics (*p* < 0.01, Figure 2A). Conversely, the miR-10a-5p inhibitor significantly reduced its expression *(p* < 0.01, Figure 2D). Subsequent qRT-PCR increases revealed that the proliferation-associated genes, including *Cyclin B*, *Cyclin D1*, cyclin-dependent kinase 4 (*CDK4*), cyclin-dependent kinase 1 (*CDK1*), antigen-identified by monoclonal antibody Ki 67 (*ki67*), and proliferating cell nuclear antigen (*PCNA*), exhibited reduced mRNA expression upon miR-10a-5p mimic treatment (*p* < 0.01, Figure 2B). In contrast, their expressions increased upon miR-10a-5p inhibition, specifically *Cyclin B*, *Cyclin D1*, *CDK4*, *CDK1*, and *ki67* (*p* < 0.01 and *p* < 0.05, Figure 2E). The CCK-8 analysis demonstrated temporal changes in preadipocyte viability post-transfection at the time points of 12, 24, 36, 48, and 60 h. The miR-10a-5p mimics hampered proliferation, while its inhibitor bolsters the same (*p* < 0.01, Figure 2C,F). Furthermore, EdU incorporation assays provided insights into the proliferation status post-transfection. Overexpression of miR-10a-5p led to a pronounced decline in EdU-positive cells, whereas its inhibitor showed an inverse effect, enhancing the ratio of EdU-positive cells (Figure 2G,H). Flow cytometric data echoed these findings. The miR-10a-5p mimics induced an increase in cell proportions at the G1 and G2 phases, as well as a concomitant decrease in the S phase. In contrast, the miR-10a-5p inhibitor exhibited the reverse effect (Figure 2I,J). Our findings underscore the potential of miR-10a-5p as a pivotal regulator in the proliferation dynamics of porcine preadipocytes.

### 3.3. miR-10a-5p Enhancement of Porcine Preadipocyte Differentiation

To determine the role of miR-10a-5p on the differentiation of porcine preadipocytes, we used qRT-PCR, western blotting, and ORO techniques. Following an 8-day transfection with either miR-10a-5p mimics or inhibitors, qRT-PCR was utilized to detect the expression levels of adipogenic markers such as peroxisome proliferator-activated receptor gamma (*PPARγ*), CCAAT enhancer-binding protein alpha (*CEBP*/*α*), and fatty acid-binding protein 4 (*aP2*). We found that miR-10a-5p mimics significantly enhanced the expression of PPARγ, CEBP/α, and aP2, while their levels were notably reduced by the miR-10a-5p inhibitor (*p* < 0.01 and *p* < 0.05, Figure 3A,B). Using western blotting, we observed that overexpression of miR-10a-5p significantly increased the protein levels of PPARγ and aP2, while miR-10a-5p inhibition notably reduced these proteins (Figure 3C,D and Appendix A). Furthermore, the lipid accumulation in the cells, a key indicator of adipogenesis, was assessed through ORO staining. The results demonstrated enhanced lipid deposition upon miR-10a-5p overexpression on the eighth day of differentiation. In contrast, the inhibition of miR-10a-5p significantly diminished the accumulation of lipid droplets (Figure 3E,F). These results highlight the integral role of miR-10a-5p in promoting the differentiation of porcine preadipocytes.

### 3.4. KLF11: A Target of miR-10a-5p in Modulating Preadipocyte Proliferation and Differentiation

To identify the downstream targets of miR-10a-5p, we utilized online prediction tools such as miRDB, Targetscan, and TangetMiner. As a result, KLF11 emerged as a potential target gene of miR-10a-5p (Figure 4A). Notably, the seed sequence of miR-10a-5p exhibited considerable conservation across several species, including humans, mice, rats, and other mammals (Figure 4B). To ascertain the direct interaction between miR-10a-5p and KLF11 3′UTR, we introduced mutations at the miR-10a-5p binding sites within the KLF11 3′UTR (Figure 4C). We also tracked the expression dynamics of KLF11 during differentiation and observed an inverse relationship with miR-10a-5p, with KLF11 expression rising as differentiation proceeded (Figure 4D). Dual-luciferase reporter assays, using both the wild-type KLF11 3′UTR (KLF11 3′UTR Wt) and its mutated counterpart (KLF11 3′UTR Mut), were established. The miR-10a-5p mimics substantially reduced the luciferase activity of the wild-type KLF11 3′UTR, an effect absent from the mutated 3′UTR (Figure 4E). Furthermore, the miR-10a-5p mimics led to a significant downregulation in KLF11 mRNA levels, whereas their inhibitors upregulated the mRNA and protein expression of KLF11 compared to their respective controls (Figure 4F,G and Appendix A). Overall, our data strongly suggests that KLF11 is a direct downstream target of miR-10a-5p.

### 3.5. KLF11: A Negative Regulator in Preadipocyte Differentiation

Overexpression of KLF11 led to distinct reductions in both the mRNA and protein levels of the adipogenic markers PPARγ and aP2 (Figure 5A,C and Appendix A). Additionally, there was a significant decrease in lipid droplet formation (Figure 5E). Conversely, when KLF11 was silenced, there was a notable increase in the mRNA and protein expression of PPARγ and aP2, yet the number of lipid droplets significantly decreased (Figure 5B,D,F and Appendix A). These findings collectively suggest that KLF11 acts as an inhibitor of preadipocyte adipogenic differentiation. To further verify the relationship between miR-10a-5p and KLF11, we conducted rescue experiments. In cells overexpressing KLF11 and transfected with miR-10a-5p mimic controls (OE-KLF11+ mimics NC group), the expression levels of adipogenic markers and lipid droplet formation in the OE-KLF11+miR-10a-5p mimics group were significantly increased (Figure 5A,C,E and Appendix A). Meanwhile, in cells with silenced KLF11 and treated with miR-10a-5p inhibitor controls (si-KLF11+inhibitor NC group), the expression of adipogenic markers and lipid droplet formation in the si-KLF11+miR-10a-5p inhibitor group was significantly decreased (Figure 5B,D,F and Appendix A). This suggests that miR-10a-5p can counteract KLF11′s inhibitory effect on adipogenic differentiation of preadipocytes.

## 4. Discussion

Fat deposition significantly influences pig growth, affecting pork quality, production efficiency, and reproductive traits [22]. These fatty traits have genetic variations. Identifying molecular markers influencing fat deposition can accelerate genetic progress. Pigs, with genetics and physiology resembling humans, are ideal models for human obesity and metabolic syndrome studies [23]. Aside from genes, non-coding RNAs such as miRNAs, lncRNAs, and circRNAs regulate preadipocyte proliferation and differentiation.

MiRNAs are pivotal in adipocyte differentiation and maturation [24,25]. PPARγ is considered the master regulator of adipocyte differentiation and is directly targeted by miR-27a, miR-27b, and miR-130 [26]. In addition, miR-27a and miR-27b have also been demonstrated to be important negative regulators of both mouse and human adipogenesis [27,28]. Recently, it was reported that the treatment of porcine pre-adipocytes with microvesicles produced from cells overexpressing miR-130b caused impaired adipocyte differentiation and repression of PPARγ [29]. Furthermore, miR-155 has been shown to directly target CREB and C/EBPβ. C/EBPβ was a critical transcription factor induced early in the process of adipogenesis [30]. Similarly, miR-138 impaired adipocyte differentiation by disrupting the activation of PPARγ by targeting EID-1 [31]. miRNAs play a critical role in regulating fat deposition and adipocyte differentiation, but many miRNA functions have not yet been studied and reported.

Thus, we delved into the molecular mechanism of miR-10a-5p in porcine fat metabolism, revealing the role of miR-10a-5p in preadipocyte differentiation by targeting the KLF11 gene. Previous research has shown miR-10a-5p’s involvement in inhibiting chicken granulosa cell proliferation by targeting MAPRE1 to suppress CDK2, as well as suppressing melanoma cell proliferation and migration [32,33]. It also impacts keratinocyte proliferation and directly targets hyaluronan synthase 3 [34]. These findings emphasize miR-10a-5p’s significant role in cell regulation. Our study further investigated its function in preadipocyte proliferation using various experiments. The results showed that miR-10a-5p mimics inhibited, while miR-10a-5p inhibitors promoted cell proliferation.

miR-10a-5p has been linked to adipocyte differentiation. RNA analysis in mice suggests its role in inflammation and differentiation in adipose macrophages and adipocyte stem cells, promoting new white adipocyte formation [15]. Previous research indicates its ability to attenuate adipogenic differentiation by targeting the 3’UTR region of Map2k6 and Fasn [16]. Map2k6 influences adipokine expression in precursor adipocytes [35], and miR-10a-5p is known to regulate genes such as Nr4a3 and Dll4 related to adipogenesis suppression [36,37]. In this study, miR-10a-5p enhanced adipogenic differentiation in porcine precursor adipocytes. However, within goat intramuscular preadipocytes, it inhibited differentiation by targeting KLF8 [38]. This discrepancy might be due to interspecies differences. Notably, our study confirmed miR-10a-5p’s binding to KLF11, affecting porcine preadipocyte differentiation.

The KLF11 gene, part of the KLFs family, was first identified in human pancreatic cells (CFPAC-1) in 1998 by the Raul Urrutia laboratory [19]. KLF11 is an important transcription factor involved in physiological processes such as cell growth, differentiation, glycolipid metabolism, and oxidative stress [39]. It is essential for modulating the genomic program in white adipocytes to induce browning [40]. Studies have shown that KLF11 is involved in sterol-responsive element-binding proteins (SREBP), mediated cholesterol production, and transport in mouse endothelial cells [41]. In mice, liver overexpression of KLF11 activates the PPARα signaling pathway, subsequently influencing lipid metabolism [21]. KLF11 can also down-regulate the expression of SREBP-1c and FAS in the mouse liver, inhibiting lipid synthesis. Furthermore, the study finds that KLF11 inhibits the differentiation of ovine SVF [17]. miR-10b-5p, which shares a seed sequence with miR-10a-5p, targets KLF11 and aids in glucose homeostasis, offering potential diabetes treatment [42]. In summary, KLF11 plays an important role in the differentiation of preadipocytes. Therefore, this study explored the effects of KLF11 on the differentiation of precursor adipocytes. The results showed that overexpression of KLF11 inhibited the differentiation of preadipocytes, while interference with KLF11 promoted the differentiation of preadipocytes. KLF11 could also hinder the promotion of miR-10a-5p in preadipocyte differentiation. Therefore, KLF11 may be necessary for the differentiation and formation of porcine preadipocytes.

## 5. Conclusions

This study demonstrates that miR-10a-5p suppresses proliferation and promotes the adipogenic differentiation of porcine precursor adipocytes. Mechanistic studies have shown that miR-10a-5p reduces KLF11 expression by binding to the 3‘UTR of KLF11, promoting adipogenesis. Our results lay a theoretical foundation for understanding the role and mechanism of miR-10a-5p in regulating adipogenic differentiation in pigs while offering valuable insights into the role of miRNAs in regulating fat deposition and enhancing meat quality.

## Figures and Tables

**Figure 1 animals-14-00337-f001:**
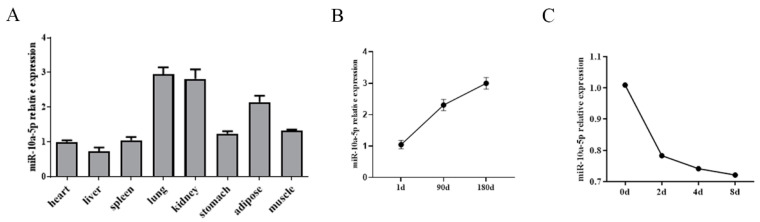
Expression patterns of miR-10a-5p. (**A**) miR-10a-5p expression profile in various tissues of pigs; (**B**) expression trend of miR-10a-5p during the subcutaneous adipose tissue development of pigs; (**C**) expression of miR-10a-5p during the differentiation of porcine preadipocytes.

**Figure 2 animals-14-00337-f002:**
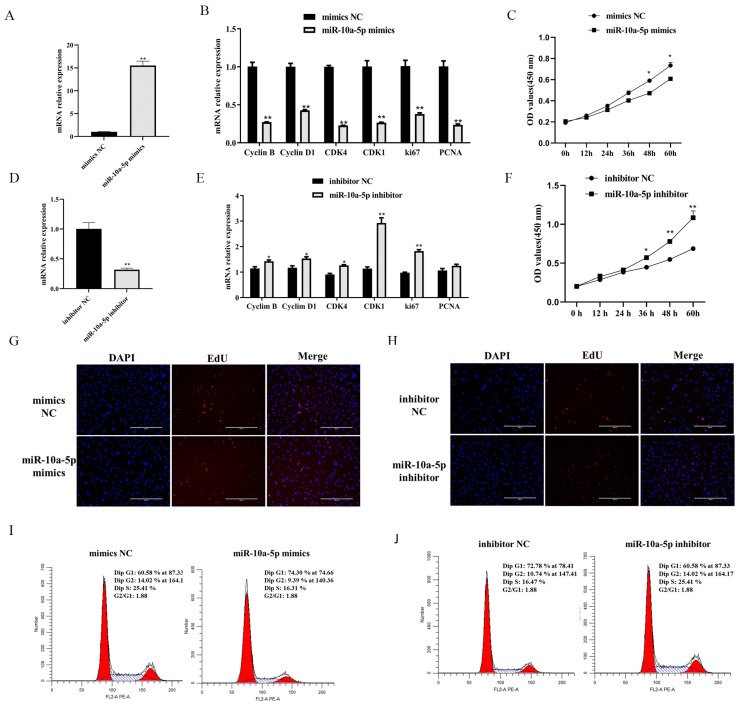
The role of miR-10a-5p in modulating porcine preadipocyte proliferation. (**A**,**D**) Transfection efficiency of miR-10a-5p mimics or inhibitors; (**B**,**E**) mRNA expression profiling of key proliferation-associated genes post-miR-10a-5p mimic or inhibitor-transfection; (**C**,**F**) temporal cell proliferation trajectories post-miR-10a-5p mimic or inhibitor-transfection, assessed via the CCK-8 assay; (**G**,**H**) EdU assays illustrating the proliferation rate of porcine preadipocytes post-transfection with either miR-10a-5p mimics or inhibitors; (**I**,**J**) flow cytometric representation of cell cycle dynamics in porcine preadipocytes following the miR-10a-5p mimic or inhibitor transfection. Significance markers: ** and * denote *p* < 0.01 and *p* < 0.05, respectively.

**Figure 3 animals-14-00337-f003:**
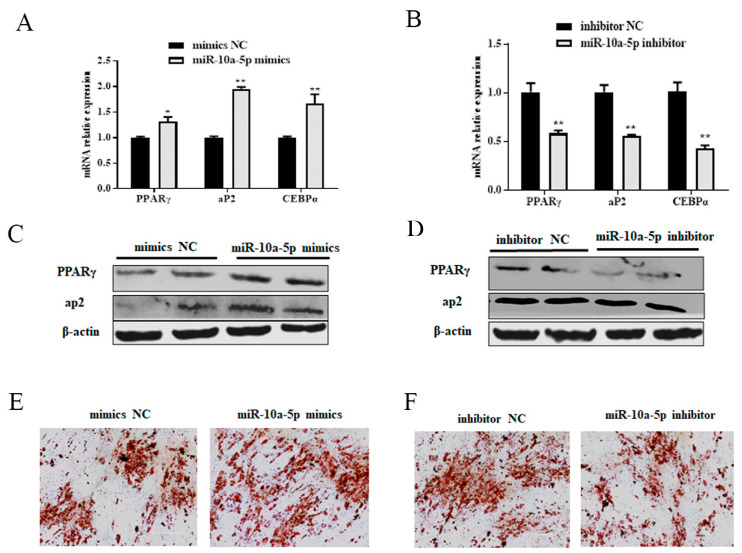
Role of miR-10a-5p in porcine preadipocyte differentiation. (**A**,**B**) mRNA expression levels of adipogenic markers in preadipocytes post-transfection with miR-10a-5p mimics or inhibitors; (**C**,**D**) protein levels of adipogenic markers in preadipocytes following miR-10a-5p mimics or inhibitor transfection; (**E**,**F**) Oil Red O staining after miR-10a-5p mimics or inhibitor treatment. Significance markers: ** and * denote *p* < 0.01 and *p* < 0.05, respectively.

**Figure 4 animals-14-00337-f004:**
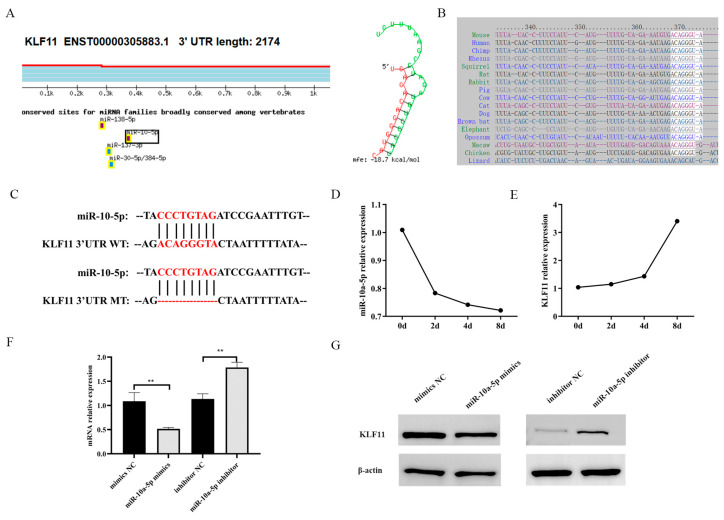
miR-10a-5p directly binds to KLF11 3′UTR. (**A**) The predicted binding alignment of the miR-10a-5p seed sequence with KLF11 3′UTR; (**B**) conservation analysis of the miR-10a-5p seed sequence across various species; (**C**) depiction of the miR-10a-5p binding site mutation within the KLF11 3′UTR; (**D**) KLF11 expression trend during porcine preadipocyte differentiation; (**E**) dual-luciferase reporter activity in cells transfected with wild-type vs. mutant KLF11 3′UTR constructs; (**F**) qRT-PCR analysis showing KLF11 mRNA expression in porcine preadipocyte post-transfection with miR-10a-5p mimics or inhibitors; (**G**) western blotting results indicating KLF11 protein expression in porcine preadipocytes following the transfection with miR-10a-5p mimics or inhibitors. ** *p* < 0.01.

**Figure 5 animals-14-00337-f005:**
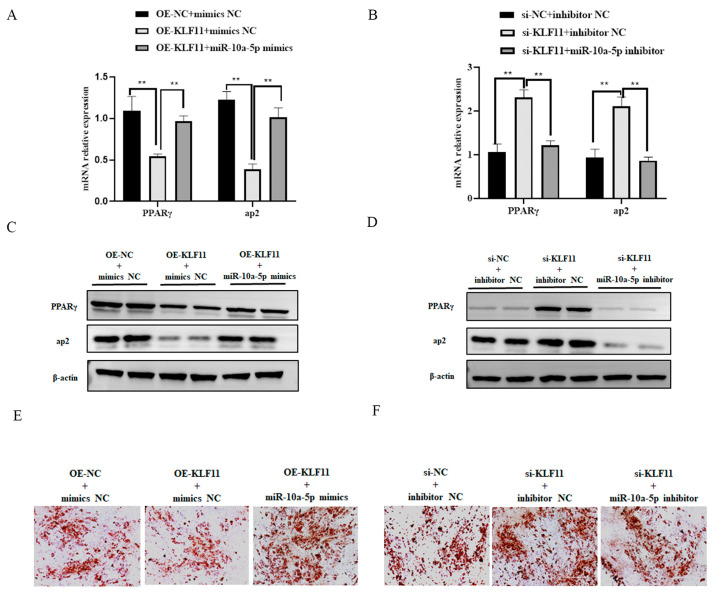
KLF11 modulates preadipocyte differentiation. (**A**,**B**) qRT-PCR analysis showing relative expression levels of adipocyte markers PPARγ, C/EBPα, and aP2; (**C**,**D**) western blot analysis revealing relative protein expression levels of PPARγ, C/EBPα, and aP2; (**E**,**F**) visualization of lipid droplet accumulation via Oil Red O staining. ** *p* < 0.01.

**Table 1 animals-14-00337-t001:** Primer sequences used for qRT-PCR.

Primer Name	Primer Sequence
*18S*-F	CCCACGGAATCGAGAAAGAG
*18S*-R	TTGACGGAAGGGCACCA
*PCNA*-F	GTGATTCCACCACCATGTTC
*PCNA*-R	TGAGACGAGTCCATGCTCG
*ki67*-F	AGCCCGTATCGTGTGCAAAA
*ki67*-R	CCTGCATCTGTGTAAGGGCA
*Cyclin D1*-F	GCGAGGAACAGAAGTGCG
*Cyclin D1*-R	TGGAGTTGTCGGTGTAGATGC
*Cyclin B*-F	TGGCTAGTGCAGGTTCAG
*Cyclin B*-R	CAGTCACAAAGGCAAAGT
*CDK1*-F	CCCTCCTGGTCAGTTCAT
*CDK1*-R	TAGGCTTCCTGGTTTCC
*CDK4*-F	GCATCCCAATGTTGTCCG
*CDK4-R*	GGGGTGCCTTGTCCAGATA
*PPARγ-F*	AGAGTATGCCAAGAACATCC
*PPARγ-R*	AGGTCGCTGTCATCTAATTC
*C/EBPα-F*	AGCCAAGAAGTCGGTAGA
*C/EBPα-R*	CGGTCATTGTCACTGGTC
*aP2-F*	AAGTCAAGAGCACCATAACC
*aP2-R*	GATACATTCCACCACCAACT
*KLF11-F*	AAGCGGCATGACAGTGAGAG
*KLF11-R* miR-10a-5p-F U6	GAGGAGTCATGCACAGAGTTG TACCCTGTAGATCCGAATTTGT AACGCTTCACGAATTTGCGT

## Data Availability

The data presented in this study are available on request from the corresponding author.

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
