# Peer review of "miR-10a-5p Regulates the Proliferation and Differentiation of Porcine Preadipocytes Targeting the KLF11 Gene"

_animals, 2024, doi:10.3390/ani14020337_

Round 1

Reviewer 1 Report

Comments and Suggestions for Authors

In this study, authors analyzed the effects of miR-10a-5p in regulating preadipocyte proliferation and differentiation. They further analyzed KLF11 gene as a target of miR-10a-5p. Experiments were properly designed and carried out. I only have minor comments:

Line 10: “employing techniques such as …. Studies.” Can be deleted. Instead, more information about the preadipocyte preparation can be added.

Fig. 4E: the activity change appeared quite small, suggesting that KLF11 might not be a good target of miR-10a-5p.

Comments on the Quality of English Language

Minor language check is needed.

Author Response

Response to Reviewer 1 Comments

Dear reviewers:

Thank you for your decision and constructive comments on my manuscript. We have carefully considered the suggestion of Reviewers and make some changes. We have tried our best to improve and made some changes in the manuscript. The yellow part that has been revised according to your comments.

Point 1: Line 10: “employing techniques such as …. Studies.” Can be deleted. Instead, more information about the preadipocyte preparation can be added.

Response 1:

Thank you for your suggestions.

We delete Abstract part of “employing techniques such as …. Studies”(line 19-20). The method of the preadipocyte preparation was described in the ‘2.2. Cell culture and differentiation induction’ (line 104-113). Please check it.

Point 2: Fig. 4E: the activity change appeared quite small, suggesting that KLF11 might not be a good target of miR-10a-5p.

Response 2:

In the Fig. 4E, the activity change appeared quite small, but it is significant differences. And we also predicted that miR-10a-5p could bind to the 3'UTR of KLF11 by various software. We also demonstrated that miR-10a-5p and KLF11 expression were in opposite trends by qRT-PCR and WB. From these results, we demonstrate that KLF11 is a target gene of miR-10a-5p.

Reviewer 2 Report

Comments and Suggestions for Authors

Comments

Abstracts should be improved to explain the functional studies performed.

Introduction

Please update the references. There some important papers on porcine miRNA and adipocyte regulation missing.

In addition, you should also consider differences between porcine intramuscular and subcutaneous fat tissue.

The direct link between miR-10a-5b and KLF11 becomes not clear in the Introduction.

Are there are other target genes and other miRNAs known for porcine adipocytes and do their pathways interfere with each other.

Line 80: Can you please add the breed and line of the piglets. How many piglets and samples were used in these experiments.

Table 1: please add the reference genome for the primer design and whether forward or reverse sequences were used.

Can you please outline the limitations of your study:

expression analysis for differentially expressed of small RNAs to find the most differentially expressed miRNAs, testing the number of differentially expressed genes and proteins

Please also discuss how to see your results in comparison to further analyses.

Figure 1A: did you also cell culture from tissues other than subcutaneous fat tissue. Please add this information in M&M and explain in the figure legend.

Figure 1B and 1C: please explain the days in these figures.

Figure 2: abbreviations only partly explained, text unreadable in the figures. May you add explanations in the legend.

Discussion and conclusions are short and should be expanded.

Pathways should be discussed and limitations because there might be further miRNAs influencing preadipocytes.

Differences to other species should be discussed.

Line 333-334: is too general, please explain and discuss this statement. Which further research is necessary to come to this conclusion.

Comments on the Quality of English Language

No comments

Author Response

Response to Reviewer 2 Comments

Dear reviewers:

Thank you for your decision and constructive comments on my manuscript. We have carefully considered the suggestion of Reviewers and make some changes. We have tried our best to improve and made some changes in the manuscript. The yellow part that has been revised according to your comments.

Point 1:

Abstracts should be improved to explain the functional studies performed.

Response 1:

Thank you for your suggestions. We have improved the Abstracts section. We delete Abstract part of “employing techniques such as …. Studies”. We have added new content (line 20-26). Please check it.

Point 2:

Please update the references. There some important papers on porcine miRNA and adipocyte regulation missing.

Response 2:

Thank you for your suggestions. We have added new content in the Introduction section, and also added some important papers on porcine miRNA and adipocyte regulation (line 51-57). Please check it.

Point 3:

In addition, you should also consider differences between porcine intramuscular and subcutaneous fat tissue.

Response 3:

Thank you for your suggestions. We agree with your view very much. Subcutaneous fat (SCF) and intramuscular fat (IMF) deposition is relevant to meat production and quality in pigs. In our study we only investigated the effect of miRNA-10a-5p on subcutaneous fat deposition, and we will follow up with a study on the effect of miRNA-10a-5p on intramuscular fat deposition.

Point 4:

The direct link between miR-10a-5b and KLF11 becomes not clear in the Introduction.

Response 4:

Thank you for your suggestions. No direct link between miR-10a-5b and KLF11 has been reported in previous studies. We added the results of this study regarding the direct link between miR-10a-5b and KLF11 in the last paragraph of the introduction section (line 79-83). Please check it.

Point 5:

Are there are other target genes and other miRNAs known for porcine adipocytes and do their pathways interfere with each other.

Response 5:

The target gene on miR-10a-5p has been studied in other species and has not been reported in pigs. Thank you for your guiding comments, which will be of great help to us in our next research.

Point 6:

Line 80: Can you please add the breed and line of the piglets. How many piglets and samples were used in these experiments.

Response 6:

Thank you for your suggestions. We added the breed and number of pigs in the Materials and Methods section (line96-103). Please check it.

“Jinfen White pigs (castrated boars) were used as experimental animals in present study, provided by Datong Pig Farm (Shanxi, China). Three 90-day-old Jinfen White pigs (castrated boars) were killed by electric shock and bloodletting. After that, longissimus dorsi muscle, heart, spleen, kidney, liver, lung, stomach and subcutaneous fat were col-lected. In addition, nine Jinfen White pigs (castrated boars) at 1, 90, and 180 days of age were selected, 3 pigs per age. After being killed by electric shock and bloodletting, subcu-taneous fat was collected. The tissues were instantly frozen in liquid nitrogen and after-ward kept in a −80 â—¦C refrigerator.”

Point 7:

Table 1: please add the reference genome for the primer design and whether forward or reverse sequences were used.

Response 7:

Thank you for your suggestions.

We added the reference genome for the primer design in ‘2.4. Quantitative real-time polymerase chain reaction’ section (line 139-140).

Forward and reverse sequences were used, and sequences are presented in Table 1 (line 141-142).

Please check it.

Point 8:

Can you please outline the limitations of your study:

expression analysis for differentially expressed of small RNAs to find the most differentially expressed miRNAs, testing the number of differentially expressed genes and proteins

Please also discuss how to see your results in comparison to further analyses.

Response 8:

Thank you for your suggestions. In this study we examined the expression pattern of miRNA-10a-5p in only one species and found that the expression of miRNA-10a-5p changed with fat deposition, which is hypothesized to function in the process of fat deposition. We should detect changes in miRNA-10a-5p expression in multiple species to clarify its function. To detect changes in gene and protein expression, we should test more marker genes and proteins to prove the accuracy of the results.

Point 9:

Figure 1A: did you also cell culture from tissues other than subcutaneous fat tissue. Please add this information in M&M and explain in the figure legend.

Response 9:

Thank you for your suggestions. We didn't cell culture from tissues other than subcutaneous fat tissue.

Point 10:

Figure 1B and 1C: please explain the days in these figures.

Response 10:

Thank you for your suggestions.

1-day-old pigs are newborn piglets. 90-day-old pigs are in the rapid growth phase. 180-day-old pigs are in a period of fat deposition.

Cells were collected once every two days during precursor adipocyte differentiation, for a total of eight days of differentiation and four collections.

Point 11:

Figure 2: abbreviations only partly explained, text unreadable in the figures. May you add explanations in the legend.

Response 11:

Thank you for your suggestions. I added the full name of all genes to ‘3.2 Inhibition of porcine preadipocyte proliferation by miR-10a-5p’ (line 227-228). Please check it.

Point 12:

Discussion and conclusions are short and should be expanded.

Pathways should be discussed and limitations because there might be further miRNAs influencing preadipocytes.

Differences to other species should be discussed.

Response 12:

Thanks. The advices are very helpful to improve the quality of this paper. We have modified both the discussion and conclusion sections (line 326-337, 361-375, 377-383). Please check it.

Point 13: Line 333-334: is too general, please explain and discuss this statement. Which further research is necessary to come to this conclusion.

Response 13:

Thank you for your suggestions. ‘Line 333-334’ have been rewritten (line 377-378). Please check it.

Reviewer 3 Report

Comments and Suggestions for Authors

Dear Authors,

The manuscript presents the molecular function of miRNA  miR-10a-5b and its relationship with KLF11 transcription factors. In mammals, the KLF11 was recognized as TF governing adipose differentiation, adipogenesis and onset of obesity. In the present study, necessary controls to confirm all experimental steps were used. The manuscript is well written. However, I have a few minor comments.

1. In the material and method section, it was said that 7-day-old piglets were used in the experiment, but how many piglets? 

2. which breeds were used? 

3. Were all cells pooled after being isolated from the neck and back?

4. Were the cells from different piglets pooled? There is a lack of this information, which makes it possible to repeat this experiment for the other researchers.

5. How many technical replicate experiments were performed?

6. In the first results section is given information about the expression pattern of investigated miRNA in different pig tissue, but in the material and method section is not given about this experiment. Please add this information: which pig tissue was collected, from which pigs, and which breeds, were the same animals in the cell culture experiment?

Author Response

Response to Reviewer 3 Comments

Dear reviewers:

Thank you for your decision and constructive comments on my manuscript. We have carefully considered the suggestion of Reviewers and make some changes. We have tried our best to improve and made some changes in the manuscript. The yellow part that has been revised according to your comments.

Point 1: In the material and method section, it was said that 7-day-old piglets were used in the experiment, but how many piglets?

Response 1:

We used 7-day-old piglets to isolate precursor adipocytes at 1- to 2-week intervals, so the exact number of piglets could not be determined.

Point 2: which breeds were used?

Response 2:

The experimental pig breed is Jinfen White Pig.

Point 3: Were all cells pooled after being isolated from the neck and back?

Response 3:

Piglets have few fat deposits, so we separated the neck and back fat, then pooled them to isolate preadipose cells.

Point4: Were the cells from different piglets pooled? There is a lack of this information, which makes it possible to repeat this experiment for the other researchers.

Response 4:

Thank you for your suggestions. Cells from different piglets were not merged, and we did replicate experiments using precursor adipocytes from different piglets.

Point5: How many technical replicate experiments were performed?

Response 5:

Thank you for your suggestions. We performed three technical replications of the experiment

Point6: In the first results section is given information about the expression pattern of investigated miRNA in different pig tissue, but in the material and method section is not given about this experiment. Please add this information: which pig tissue was collected, from which pigs, and which breeds, were the same animals in the cell culture experiment?

Response 6:

Thanks. The advices are very helpful to improve the quality of this paper. We added the breed and number of pigs in the Materials and Methods section (line 96-103). Please check it.

“Jinfen White pigs (castrated boars) were used as experimental animals in present study, provided by Datong Pig Farm (Shanxi, China). Three 90-day-old Jinfen White pigs (castrated boars) were killed by electric shock and bloodletting. After that, longissimus dorsi muscle, heart, spleen, kidney, liver, lung, stomach and subcutaneous fat were col-lected. In addition, nine Jinfen White pigs (castrated boars) at 1, 90, and 180 days of age were selected, 3 pigs per age. After being killed by electric shock and bloodletting, subcu-taneous fat was collected. The tissues were instantly frozen in liquid nitrogen and after-ward kept in a −80 â—¦C refrigerator.”

Reviewer 4 Report

Comments and Suggestions for Authors

This manuscript investigates the regulatory effect of miR-10a-5b on the proliferation and differentiation of porcine preadipocytes. Furthermore, miR-10a-5b could target KLF11 and suppress its mRNA and protein expression, hence regulating the proliferation and differentiation of preadipocytes, which is critical for functional miRNAs in fat deposition. The manuscript is valuable enough to be published with minor revisions.

Simple Summary

The Simple Summary section should be clearer and more concise.

Introduction

The introduction should explain why this miRNA is being studied and where it was discovered.

Materials and Methods

Line 80-81 Subcutaneous adipose tissue from the back and neck of 7-day-old piglets was harvested and transferred to sterile Petri dishes. Whether the cells utilized in the test were derived from back or neck tissue?

Discussion

The discussion is a little superficial and could use some polishing. And the discussion should include a discussion of the effects of KLF11 on adipocyte proliferation and differentiation.

Comments on the Quality of English Language

no

Author Response

Response to Reviewer 4 Comments

Dear reviewers:

Thank you for your decision and constructive comments on my manuscript. We have carefully considered the suggestion of Reviewers and make some changes. We have tried our best to improve and made some changes in the manuscript. The yellow part that has been revised according to your comments.

Point 1:

Simple Summary

The Simple Summary section should be clearer and more concise.

Response 1:

Thank you for your suggestions. We have rewritten the Simple Summary section ( line10-14). Please check it.

‘The aim of this study was to explore the regulatory effect of miR-10a-5b on proliferation and differentiation of porcine preadipocytes. Our results showed that miR-10a-5b could target KLF11 suppress proliferation and promotes differentiation in preadipocytes. Therefore, we concluded that miR-10a-5b plays a significant role in regulating the proliferation and differentiation of preadipocyte.’

Point 2:

Introduction

The introduction should explain why this miRNA is being studied and where it was discovered.

Response 2:

Thanks. The advices are very helpful to improve the quality of this paper. We have rewritten the third paragraph of the introductory section. We explain in the introduction why we studied miR-10a-5b in preadipose cells (line 58-66). Please check it.

Point 3:

Materials and Methods

Line 80-81 “Subcutaneous adipose tissue from the back and neck of 7-day-old piglets was harvested and transferred to sterile Petri dishes.” Whether the cells utilized in the test were derived from back or neck tissue?

Response 3:

Thank you for your suggestions. Piglets have few fat deposits, so we separated the neck and back fat, then pooled them to isolate preadipose cells.

Point4:

Discussion

The discussion is a little superficial and could use some polishing. And the discussion should include a discussion of the effects of KLF11 on adipocyte proliferation and differentiation.

Response 4:

Thanks. The advices are very helpful to improve the quality of this paper. We added the discussion part about the effects of KLF11 on adipocyte proliferation and differentiation (line 361-375). Please check it.